# Progressively Stacking 2.0: A Multi-stage Layerwise Training Method for BERT Training Speedup

## Abstract

Pre-trained language models, such as BERT, have achieved significant accuracy gain in many natural language processing tasks. Despite its effectiveness, the huge number of parameters makes training a BERT model computationally very challenging. In this paper, we propose an efficient multi-stage layerwise training (MSLT) approach to reduce the training time of BERT. We decompose the whole training process into several stages. The training is started from a small model with only a few encoder layers and we gradually increase the depth of the model by adding new encoder layers. At each stage, we only train the top (near the output layer) few encoder layers which are newly added. The parameters of the other layers which have been trained in the previous stages will not be updated in the current stage. In BERT training, the backward computation is much more time-consuming than the forward computation, especially in the distributed training setting in which the backward computation time further includes the communication time for gradient synchronization. In the proposed training strategy, only top few layers participate in backward computation, while most layers only participate in forward computation. Hence both the computation and communication efficiencies are greatly improved. Experimental results show that the proposed method can achieve more than $110\%$ training speedup without significant performance degradation.

## 1 Introduction

In recent years, the pre-trained language models, such as BERT (Devlin et al., 2018), XLNet (Yang et al., 2019), GPT (Radford et al., 2018), have shown their powerful performance in various areas, especially in the field of natural language processing (NLP). By pre-trained on unlabeled datasets and fine-tuned on small downstream labeled datasets for specific tasks, BERT achieved significant breakthroughs in eleven NLP tasks (Devlin et al., 2018). Due to its success, a lot of variants of BERT were proposed, such as RoBERTa (Liu et al., 2019b), ALBERT (Lan et al., 2019), Structbert (Wang et al., 2019) etc., most of which yielded new state-of-the-art results.

Despite the accuracy gains, these models usually involve a large number of parameters (e.g. BERT-Base has more than 110M parameters and BERT-Large has more than 340M parameters), and they are generally trained on large-scale datasets. Hence, training these models is quite time-consuming and requires a lot of computing and storage resources. Even training a BERT-Base model costs at least $7k$ (Strubell et al., 2019), let alone the other larger models, such as BERT-Large. Such a high cost is not affordable for many researchers and institutions. Therefore, improving the training efficiency should be a critical issue to make BERT more practical.

Some pioneering attempts have been made to accelerate the training of BERT. You et al. (2019) proposed a layerwise adaptive large batch optimization method (LAMB), which is able to train a BERT model in 76 minutes. However, the tens of times speedup is based on the huge amount of computing and storage resources, which is unavailable for common users. Lan et al. (2019) proposed an ALBERT model, which shares parameters across all the hidden layers, so the memory consumption is greatly reduced and training speed is also improved due to less communication overhead. Gong et al. (2019) proposed a progressively stacking method, which trains a deep BERT

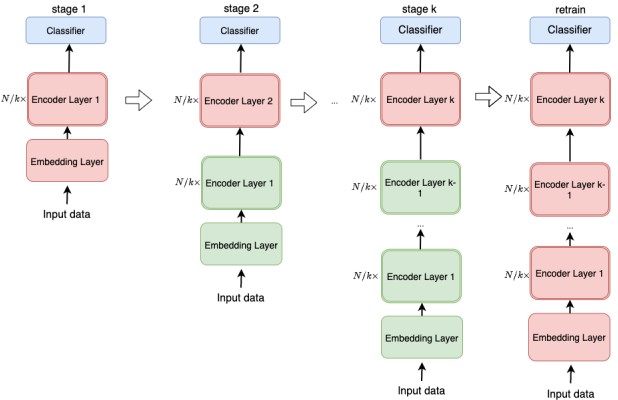

Figure 1: The framework of MSLT method. The green blocks mean that these layers only participate in forward computation and are not updated. The red blocks mean that these layers are trained in this stage and they participate in both forward and backward computations.

network by progressively stacking from a shallow one. Utilizing the similarity of the attention distributions across different layers, such a strategy achieves about $25\%$ speedup without significant performance loss.

Progressively stacking provides a novel training strategy, namely training a BERT model from shallow to deep. However, progressively stacking only has a high training efficiency at the initial stage in which the model depth is small. As the training goes on, the model depth increases and the training speed decreases. The low efficiency of the later stages makes the overall speedup of progressively stacking limited. Note that in the progressively stacking method, the bottom layers are trained with longer time than the top layers. However, we observe that though the bottom layers are updated all the time, they do not have significant changes in the later stages, in terms of the attention distribution which can reflect the functionality of the encoder layers to some extent (Gong et al., 2019). In other words, most optimization of the bottom layers has been finished in the early stage when the model is shallow. Motivated by this observation, in this work, we propose a novel multi-stage layerwise training (MSLT) approach, which can greatly improve the training efficiency of BERT. We decompose the training process of BERT into several stages, as shown in Fig. 1. We start the training from a small BERT model with only a few encoder layers and gradually add new encoder layers. At each stage (except the first stage), only the output layer and the newly added top encoder layers are updated, while the other layers which have been trained in the previous stages will be fixed in the current stage. After all the encoder layers are trained, to make the network better behaved, we further retrain the model by updating all the layers together. Since the whole model has already been well trained, this stage only requires a few steps (accounting for about $20\%$ of the total steps). Compared with the progressively stacking method, which requires a lot of steps (accounting for about $70\%$ of the total steps (Gong et al., 2019)) to train the whole model, our method is much more time-efficient.

Experimental results demonstrate the effectiveness and efficiency of the proposed method in two aspects: 1) with the same data throughput (same training steps), our method can achieve comparable performance, compared with the original training method, but consumes much less training time; 2) with the same training time, our method can achieve better performance than the original method. According to the results, the proposed method achieves more than $110\%$ speedup without significant performance degradation.

To avoid misunderstanding, it should be mentioned that some widely-known methods such as model compression (Han et al., 2015a;b) and knowledge distillation (Yim et al., 2017; Hinton et al., 2015; Sanh et al., 2019) are designed for network speedup in the inference phase. Namely, these methods are used after the model has been trained. While in this paper, we focus on the model training speedup.

## 2    RELATED WORK

Based on the bidirectional Transformer (Vaswani et al., 2017) encoder, BERT has shown its great representational ability and it achieved state-of-the-art results in eleven NLP tasks. Following BERT, many pre-trained models were proposed, such as RoBERTa (Liu et al., 2019b), XLNet (Yang et al., 2019), KBERT (Liu et al., 2019a), StructBERT (Wang et al., 2019), and so on. Higher accuracy were achieved by these models with more training data, more training steps, or more effective loss functions. However, the BERT models are generally large-scale and they need to be trained on massive datasets (e.g. BERT-base is trained on BooksCorpus and Wikipedia with totally 3.3 billion word corpus). Hence, training a BERT model is challenging in terms of both computation and storage. In the literature, some approaches were proposed to improve the training speed of BERT.

### 2.1    DISTRIBUTED TRAINING WITH LARGE BATCH SIZE

A direct way to reduce the training time is to increase the training batch size by using more machines and train the model in a distributed manner. However, traditional stochastic gradient descent (SGD) based optimization methods perform poorly in large mini-batches training. Naively increasing the batch size leads to performance degradation and computational benefits reduction (You et al., 2019). An efficient layerwise adaptive large batch optimization technique named LAMB was proposed in You et al. (2019) to address this problem. It allows the BERT model to be trained with extremely large batch size without any performance degradation. By using 1024 TPUv3 chips, LAMB reduced the BERT training time from 3 days to 76 minutes. Though tens of times speedup is achieved, these methods require a huge amount of computing and storage resources, which are far beyond the reach of common users.

### 2.2    ALBERT

ALBERT (Lan et al., 2019) adopted two parameter reduction techniques, namely factorized embedding parameterization and cross-layer parameter sharing, which significantly reduced the model size. In addition, ALBERT adopted the sentence-order prediction loss instead of the next-sentence prediction loss during pre-training, which is demonstrated to be more effective in terms of downstream performance. Since the communication overhead is directly proportional to the number of parameters in the model, ALBERT also improved the communication efficiency in distributed training setting. However, since ALBERT has almost the same computational complexity as BERT, training an ALBERT model is still very time-consuming.

### 2.3    PROGRESSIVELY STACKING

The most related work should be progressively stacking (Gong et al., 2019), which is mainly based on the observation that in a trained BERT model, the attention distributions of many heads from top layers are quite similar to the attention distributions of the corresponding heads from the bottom layers, as shown in Fig. 2. Such a phenomenon implies that the encoder layers in the BERT model have similar functionalities. Utilizing the natural similarity characteristic, to train a $N-$layer BERT model, the progressively stacking method first trains a $N/2-$layer model, and then sticks it into $N-$layer by copying the parameters of the trained $N/2$ layers. After the $N-$layer model is constructed, the progressively stacking method continues to train the whole model by updating all the parameters together. By repeatedly using such a strategy, the deep BERT model can be trained more efficiently. According to the results shown in Gong et al. (2019), progressively stacking can achieve the training time about $25\%$ shorter than the original training method (Devlin et al., 2018).

However, we can see that the speedup of progressively stacking mainly comes from the initial stage in which the model depth is small. As the model depth increases in the later stages, the training efficiency also decreases, and according to Gong et al. (2019), to guarantee the performance, more training steps should be assigned in the later stages for training the deep model. Hence, the overall speedup brought by progressively stacking is limited. Such an issue is addressed in this paper. In our work, we also train the BERT model from shallow to deep. In contrast , at each stage, we only train the top few layers and we almost keep a high training efficiency during the whole training process. Hence, much more significant speedup is achieved.

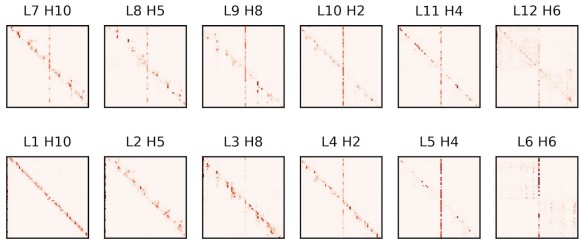

Figure 2: The attention distribution of a trained BERT-Base model given a sample sentence. We compare the attention distribution of the same heads from different layers. For example, "L5 H4" denotes the attention distribution from the forth heads of the fifth layer.

## 3 METHODOLOGY

In this section, we propose an efficient training method to accelerate the training of BERT model.

### 3.1 MOTIVATION

The large depth should be one of the main reasons making the BERT training time-consuming. The original method (Devlin et al., 2018) trains all the encoder layers simultaneously. At each training step, the parameters need to wait for the cost function gradients to propagate backwards across all the layers before update, which is very inefficient, especially when the model is very deep.

Inspired by progressively stacking (Gong et al., 2019), we also consider to train the BERT model from shallow to deep. The main problem of progressively stacking is that its training efficiency decreases as the training goes on. We observe that in the progressively stacking strategy, the bottom layers are trained for longer time than the top layers. For example, the first encoder layer (near the input layer) is updated from beginning to end, while the last encoder layer (near the output layer) is only trained at the last stage. We doubt whether it is necessary to spend much more time training the bottom layers, since some research implies that the top encoder layers play a much more significant role (Khetan & Karnin, 2020).

In BERT model, the encoder layers are mainly used to learn the dependencies of the input elements, which can be reflected by the attention distribution. In Gong et al. (2019), the authors showed that the distributions of most attention heads are mixtures of two distributions: one distribution focuses on local positions, and another focuses on the first CLS token. In addition, the authors also observed that the attention distribution of the top layers is very similar to that of the bottom layers. Using a similar way, we also visualize some important attention distributions and we get some new findings when using the progressively stacking method to train a 12-layer BERT-Base model. Specifically, we first train a 6-layer model. Then we stack the trained 6-layer model into a 12-layer model and continue to train the whole model until convergence. The attention distributions of the top 6 encoder layers of the final 12-layer BERT-Base model are shown in the first row of Fig. 3. For each layer, we randomly choose an attention head. Then we also show the attention distributions of the corresponding heads of the bottom 6 encoder layers in the second row of Fig. 3. Further, we show the attention distributions of the corresponding heads of the trained 6-layer BERT model before stacking in the third row. As a comparison, we also train a 12-layer BERT-Base model from scratch using the original method, where the parameters of the bottom 6 encoder layers use the same initialization as the above BERT model trained by progressively stacking. The forth row of Fig. 3 shows the attention distributions of the bottom 6 encoder layers of the original BERT-Base model.

Combined with Fig. 2, we find that:

**1.** Except the two obvious distributions found by Gong et al. (2019), namely the distribution focusing on local positions and the distribution focusing on the CLS token, the attention distributions of many heads also focus on the SEP token (for example, the dark vertical line in "L11 H4" in Fig. 2 corresponds to the position of SEP).

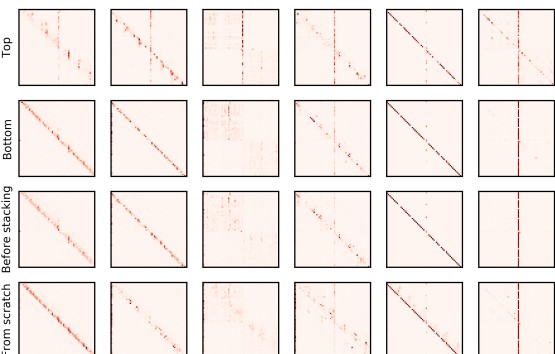

Figure 3: The first row is the attention distributions for a randomly chosen sample sentence on random 6 heads of the top 6 layers from the final trained 12-layer BERT-Base model. The second row is the attention distributions of the bottom 6 layers from the final trained 12-layer BERT-Base model. The third row is the attention distributions of the encoder layers from the trained 6-layer model before stacking. The fourth row is the attention distributions of the bottom 6 layers from the BERT-Base model trained from scratch using the original method.

**2.** Compared with the first and second rows of Fig. 3, one can see that for a trained 12-layer BERT model, some bottom layers have similar attention distributions to the corresponding top layers, which is in line with the observation in Gong et al. (2019). In addition, there are also some bottom-top layer pairs whose attention distributions are very different. On the other hand, compared with the second and third rows of Fig. 3, we can see that the attention distributions of the bottom layers from the final 12-layer model are almost the same as those of the corresponding layers from the trained 6-layer model before stacking, which implies that the further training of the bottom layers after stacking does not bring substantial optimization. The performance of the bottom encoder layers are not further improved, in terms of catching the elements' dependencies. Compared with the third and forth rows of Fig. 3, we see that the attention distributions of the trained 6-layer model are also very similar to those of the bottom layers of the BERT-Base model with all the layers jointly trained from scratch.

Therefore, it is not worth spending too much time training the bottom layers, especially updating the bottom layers is generally much more expensive than updating the top layers, since backward computation is top-down. An intuitive idea is that at each stage, let the bottom layers having been trained in the previous stages only participate in the forward computation, and only the newly added top layers as well as the output layer participate in the backward computation. Then the gradient information of the parameters from the bottom layers will not be computed and also will not be communicated in distributed training setting. So the time of both computation and communication can be greatly reduced.

## 3.2    Multi-stage Layerwise Training

Now we present our MSLT training method. Based on the above observations, to train a deep BERT model with $N$ encoder layers, we decompose the whole training process into $k$ stages, as shown in Fig. 1. We start our training from a shallow model with only $N/k$ encoder layers and we gradually add new encoder layers on the top of the model (below the output layer). At each time, we only add $N/k$ new layers. Except the first stage in which we update all the parameters, in the other stages, only the parameters of the newly added top layers and the output layer are updated. Namely, the bottom layers which have been trained in the previous stages only participate in the forward computation to provide the training input for the top layers. In addition, in the BERT model, the word embedding matrices in the input and output layers are shared. So we only update the word embedding matrix in the first stage, and in the later stages the word embedding matrix is fixed.

Similar to Gong et al. (2019), at each stage, we initialize the parameters of the newly added $N/k$ layers by copying the parameters of the top $N/k$ layers of the model trained in the previous stage. To make the final model better behaved, after each layer is trained, we further retrain the model by updating all the parameters together for a few steps.

In the BERT model training, backward computation generally takes much longer time than the forward computation, especially in the distributed training setting in which the backward computation also includes gradient synchronization. For example, in our experiment, when using the original method (Devlin et al., 2018) to train a BERT-Base model, the time of backward computation (including communication time for gradient synchronization in the distributed setting) is almost six times as the time of forward computation. The proposed MSLT method can greatly reduce the backward computation, since only top few layers participate in the backward computation in the whole training process, except the final retraining stage. Hence, the total training time will be much shorter.

### 3.3 EXTENDING TO ALBERT

The proposed MSLT strategy can also be used to speed up the training of ALBERT Lan et al. (2019). The only problem is that ALBERT shares the parameters across all the encoder layers, so we are not able to only update the top layers while keeping the other layers fixed. Hence, before applying MSLT, we make a slight modification of ALBERT. We decompose the whole encoder layers into $k$ groups and we share the parameters of the encoder layers in the same group. Then we use a similar strategy as shown in last subsection to efficiently train the ALBERT model. Specifically, at each stage, we add a new group of encoder layers and only the newly added group of layers will be updated in this stage.

Compared with the original ALBERT model, the modified ALBERT model requires more storage resources since it involves more parameters. However, the training speed is significantly improved. In real applications, one should decompose the encoder layers into a suitable number of groups to achieve a computation-storage efficiency balance, according to actual demand.

## 4 EXPERIMENT

### 4.1 EXPERIMENTAL SETUP

In this section, we perform experiments to demonstrate the effectiveness and efficiency of the propose method. Following the setup in Devlin et al. (2018), we use English Wikipedia (2,500M words) and BookCorpus (800M words) for pre-training. Though some more effective objectives were proposed in Lan et al. (2019); Steinley (2006), to make the comparison as meaningful as possible, in this experiment we still adopt the same Masked Language Model (MLM) and Next Sentence Prediction (NSP) objectives used by the original BERT (Devlin et al., 2018). We set the maximum length of each input sequence to be 128. We focus on the training time speed up of the proposed method and whether similar performance is achieved compared with the original BERT with the same setting. The models considered in this section are BERT-Base and BERT-Large, whose hyperparameters setting can be seen in Devlin et al. (2018). For each model, we decompose the training process into $k = 4$ stages. So at each stage, we train 3 encoder layers for BERT-Base, and 6 encoder layers for BERT-Large. All the experiments are performed on a distributed computing cluster consisting of 32 Telsa V100 GPU cards with 32G memory, and the batch size is 32*32=1024. We use the LAMB optimizer with learning rate 0.00088 (You et al., 2019) . All the other settings are the same as Devlin et al. (2018), including the data pre-processing, unless otherwise specified.

### 4.2 COMPARISON FOR TRAINING WITH SAME STEPS

We train both BERT-Base and BERT-Large models using the MSLT method for totally 1,000,000 steps. Specifically, each stage uses 200,000 steps, and the final model is retrained for the remaining 200,000 steps. We train a BERT-Base and a BERT-Large model with 1,000,000 steps from scratch using the original method (Devlin et al., 2018) as the baselines. For each model, the first 10,000 training steps are used for learning rate warmup. We first show the pre-training loss of all the models in Fig. 4. We can see that for both BERT-Base and BERT-Large, the loss of our method decreases faster than the baseline, and the final convergence value of our method is close to that of the baseline.

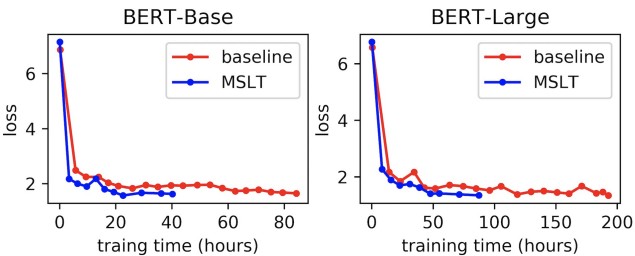

Figure 4: Pre-training loss curves of the BERT models.

Table 1: Dev set results of SQuAD and selected GLUE tasks for the pre-trained models. F1/EM scores are reported for SQuAD tasks. F1 scores are reported for QQP, MRPC, and Spearman correlations are reported for STS-B. The accuracy scores are reported for the other tasks. The result of MNLI is the average of the scores of MNLI-m and MNLI-mm tasks. The Avg result is the average of the scores of the downstream tasks to its left, where the F1 and EM scores of SQuAD tasks are firstly averaged.

| Model | time | MNLI | QNLI | SST-2 | MRPC | SQuAD1.1 | SQuAD2.0 | Avg |
|---|---|---|---|---|---|---|---|---|
| BERT-Base (ours) | 40h | 84.3 | 91.2 | 92.2 | 91.2 | 89.2/82.3 | 77.8/74.1 | 86.8 |
| BERT-Base (original) | 85h | 84.4 | 91.5 | 92.7 | 89.7 | 89.5/82.7 | 77.7/73.9 | 86.7 |
| BERT-Base (stacking) | 63h | 84.6 | 91.6 | 92.1 | 91.2 | 89.4/82.5 | 77.3/74.2 | 86.9 |
| BERT-Large (ours) | 84h | 86.2 | 92.6 | 93.2 | 92.2 | 91.6/84.8 | 81.6/79.2 | 88.8 |
| BERT-Large (original) | 188h | 86.5 | 93.0 | 93.6 | 91.7 | 91.9/85.2 | 81.5/79.1 | 88.9 |
| BERT-Large (stacking) | 136h | 86.7 | 93.1 | 93.8 | 92.4 | 92.1/85.2 | 81.7/79.2 | 89.2 |

Finally, our method achieves more than $110\%$ speedup (saves about $55\%$ training time), which is a significant improvement compared with the $25\%$ speedup achieved by progressively stacking (Gong et al., 2019).

Then we further evaluate the above models on the widely used General Language Understanding Evaluation (GLUE (Wang et al., 2018)) benchmark and the Stanford Question Answering Dataset (SQuAD1.1 and SQuAD2.0). The sequence length of the SQuAD tasks is 384. So for the SQuAD tasks, we train the last $10\%$ of the steps using the sequence of 512 to learn the positional embeddings. To better show the advantage of the proposed method, we also add the progressively stacking method Gong et al. (2019) as the baseline. Table 1 shows the Dev set results on SQuAD and selected GLUE tasks. Similar to Liu et al. (2019b), all the results are the median of five runs. For each GLUE task, we fine-tune the model with batch size 32 and we perform a grid search on the learning rate set [5e-5, 4e-5, 3e-5, 2e-5]. Following Clark et al. (2020), we fine-tune the model with 10 epochs for STS and RTE, and 3 epochs for all the other tasks. For SQuAD1.1 task, we fine-tune the model with batch size 12, epoch 2, and learning rate 3e-5. For SQuAD2.0 task, we fine-tune the model with batch size 48, epoch 2, and learning rate 5e-5.

From Table 1, we can see that for both BERT-Base and BERT-Large, the results of our method are close to those of the original BERT. In addition, the results of the baselines are comparable to those shown in Devlin et al. (2018), which confirms the validity of our proposed method.

## 4.3 COMPARISON FOR TRAINING WITH SAME TIME

In this section, we compare the performance of all the models trained with the same time. Training a BERT-Base model with 1M steps using MLST method requires about 40 hours, which is similar to the time of training a BERT-Base model using the original method for 480,000 steps (about 41 hours). In addition, we further train two BERT-Large models using MSLT with 500,000 steps (about 42 hours) and the original method with 230,000 steps (about 42 hours), respectively. The results on selected GLUE tasks are shown in Table 2. We see that with the same training time, the models trained by our method perform much better than the baselines.

Table 2: Dev set results of selected GLUE tasks for the pre-trained models by controlling training time.

| Model | steps | time | MNLI | QNLI | SST-2 | STS-B | MRPC | RTE | Avg |
|---|---|---|---|---|---|---|---|---|---|
| BERT-Base (ours) | 1M | 40h | 84.3 | 91.2 | 92.2 | 89.5 | 91.2 | 69.3 | 86.3 |
| BERT-Base (original) | 480k | 41h | 83.3 | 90.6 | 91.2 | 89.2 | 89.2 | 68.9 | 85.4 |
| BERT-Large (ours) | 500k | 42h | 84.7 | 91.5 | 92.1 | 89.9 | 91.6 | 70.7 | 86.8 |
| BERT-Large (original) | 230k | 42h | 84.1 | 90.8 | 91.1 | 89.2 | 88.9 | 68.5 | 85.4 |

Table 3: Dev set results of selected GLUE tasks for ALBERT-Base models.

| Model | time | parameters | MNLI | QNLI | SST-2 | STS-B | MRPC | Avg |
|---|---|---|---|---|---|---|---|---|
| ALBERT-Base (ours) | 32h | 32M | 82.4 | 89.9 | 90.9 | 88.8 | 90.7 | 88.5 |
| ALBERT-Base (original) | 62h | 12M | 81.8 | 89.2 | 89.4 | 88.1 | 87.4 | 87.2 |

## 4.4 ACCELERATE TRAINING OF ALBERT

As shown in Section 3.3, the proposed MSLT method can also be used to speed up the training of ALBERT. Table 3 reports the results on select GLUE tasks of the original ALBERT-Base model and the modified ALBERT-Base model trained by MSLT. All the models are trained for 1M steps and the other settings are the same as Section 4.2. We can see that the modified ALBERT model trained by MSLT achieves better performance than the original BERT. That is because the modified ALBERT model involves more parameters. Hence, the original ALBERT has higher memory efficiency while the modified ALBERT trained using MSLT has higher time efficiency and better performance.

## 4.5 EFFECT OF JOINTLY RETRAINING

In the above examples, we left 200,000 steps for jointly retraining all the layers. Here we investigate the impact of the retraining stage. Table 4 shows the results of BERT models with/without retraining. The results implies that the retraining stage can further improve the performance of the BERT model. The training efficiency of the retraining stage is much lower than the previous stages, since in this stage all the parameters are updated. However, since the model is already near-optimal after training by previous stage, the retraining stage only requires a few steps. In practice, we use $10\% \sim 20\%$ of the total steps for retraining, so the retraining stage will not make the whole training very time-consuming.

## 4.6 ONLINE TEST RESULTS ON GLUE

Lastly, we report the online test results on GLUE tasks (except WNLI) of all the models after fine-tuning, which are shown in Table 5. All the submitted models are pre-trained with 1,000,000 steps. Same as Devlin et al. (2018), for each task, we select the best fine-tuning learning rate on the Dev set.

## 5 DISCUSSION

In last section, we empirically showed the effectiveness and efficiency of the proposed MSLT method. In this section we discuss how does it work and why it can improve the convergence speed.

### 5.1 RELATIONSHIP BETWEEN MSLT AND ALTERNATING OPTIMIZATION

At first glance, MSLT is an extension of the progressively stacking method, and the main difference between MSLT and progressively stacking is that at each time progressively stacking updates all the parameters, while MSLT freezes most parameters and only updates the parameters from the few top layers. Such a strategy is similar to alternating optimization Bezdek & Hathaway (2003), which is widely used for solving multi-variable nonconvex optimization problem, due to its simple

Table 4: Results on selected GLUE tasks of BERT models with/without retraining.

| Model | steps | time | MNLI | QNLI | SST-2 | STS-B | MRPC | Avg |
|---|---|---|---|---|---|---|---|---|
| BERT-Base without retraining | 800K | 26h | 82.1 | 88.7 | 90.2 | 87.9 | 89.5 | 87.7 |
| BERT-Base with retraining | 1M | 40h | 84.4 | 91.5 | 92.7 | 89.7 | 89.7 | 89.6 |
| BERT-Large without retraining | 800K | 54h | 83.3 | 89.7 | 91.2 | 88.2 | 89.9 | 88.5 |
| BERT-Large with retraining | 1M | 84h | 86.2 | 92.6 | 93.2 | 90.2 | 92.2 | 90.9 |

Table 5: Online test results of GLUE tasks for the pre-trained models.

| Model | MNLI | QQP | QNLI | SST-2 | CoLA | STS-B | MRPC | RTE | Avg |
|---|---|---|---|---|---|---|---|---|---|
| BERT-Base (ours) | 83.9 | 71.0 | 90.4 | 93.0 | 56.6 | 84.1 | 89.2 | 68.0 | 79.5 |
| BERT-Base (baseline) | 84.2 | 71.4 | 90.7 | 93.5 | 57.2 | 84.4 | 88.8 | 68.1 | 79.8 |
| BERT-Large (ours) | 85.9 | 71.9 | 92.4 | 94.1 | 60.5 | 85.8 | 89.9 | 69.8 | 81.3 |
| BERT-Large(baseline) | 86.4 | 72.1 | 92.8 | 94.7 | 60.6 | 86.1 | 89.1 | 70.2 | 81.5 |

implementation, fast convergence, and superb empirical performance Li et al. (2019). Alternating optimization addresses the multi-variable optimization problem by updating a small set of variables and keeping the other variables fixed. The alternating optimization method is quite suitable for dealing with the problems in which updating a part of variables is much easier than updating all variables Bezdek & Hathaway (2002).

According to alternating optimization, in the retraining stage, we should update the bottom layers again and keep the other layers fixed. However, when the model is stacked deep, to compute the gradient of the parameters of the bottom layers, we have to compute the gradient of the parameters of the top layers, according to the chain rule. In this situation, updating the parameters of the bottom layers is almost as expensive as updating all the parameters. So we update all the parameters simultaneously in the retraining stage.

Overall, in the MSLT method, we utilize the advantage of alternating optimization to quickly reach a near-optimal state. When the model is deep enough, alternating optimization does not have efficiency advantage, then we use joint descent to sufficiently utilize the gradient information.

## 5.2 AVOID CONTRADICTION OF CHOICE OF LEARNING RATE

In Section 2.3, we have shown that, though the bottom layers are updated for most steps in progressively stacking method, they do not have significant change in the later stage. So it is not worthwhile to spend so much the computation to updating the parameters of bottom layers. Another issue is that for the adaptive optimizer, such as Adam or LAMB, when the variable is close to the optimal value, we should use a small learning rate, and when the variable is far from optimal, we should use a large learning rate. In progressively stacking strategy, the bottom layers are near-optimal while the newly added top layers are far from optimal since they are not trained. So they have completely different requirements for the learning rate. The MSLT method address the learning rate contradiction by first optimizing the untrained top layers and keep the trained bottom layers fixed. When all the layers are near-optimal, we further retrain all the layers with a small learning rate. Meanwhile, such a manner saves a lot of computation.

## 6 CONCLUSION

In this paper, we propose an efficient multi-stage layerwise training method for accelerating the training process of BERT. We decompose the whole training process into several stages and we adopt the progressively stacking strategy that trains a BERT model from shallow to deep by gradually adding new encoder layers. We find that the attention distributions of the bottom layers tend to be fixed early, and further training in the later stages does not bring significant changes. So in our method, at each stage, we only train the top few layers which are newly added, while the bottom layers only participate in the forward computation. Experimental results show that the proposed training method achieves significant speedup without significant performance loss, compared with the existing training method.

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
