# OpenReview forum: "Progressively Stacking 2.0: A Multi-stage Layerwise Training Method for BERT Training Speedup"
_ICLR.cc/2021/Conference — Reject_

### Official Review · AnonReviewer2 · 2020-10-27
**simple method to speedup BERT training**

**Rating:** 6
**Confidence:** 4

**Review:**

### Summary
This paper proposes a simple method, ie multi-stage layerwise training (MSLT), to speedup BERT training. Specifically, the authors progressively stack layers. The bottom layers are fixed and only the new added top layers are trained. The proposed method can achieve more than 110% training speedup and achieve comparable (slightly worse) performance. Although compared with other speedup training methods like ELECTRA [1], the idea of this paper is not novel. However, the proposed method is simple and effective to some extend. Thus, I give a marginal score to this paper.

### Strengths
* The proposed training method is simple and easy to implement.
* It can achieve 110% training speedup without significant performance degradation.

### Weaknesses and Questions
* The most related method to this paper is [2]. Thus, it is better to give more comparisons and discussions in the Experiment, so that the author can know the advantages of MSLT compared with [2].
* From Figure 4, it seems that BERT-base (baseline) needs to train around 80 hours. If training with MSLT using the same time as baselines (like train 2M steps), how about the performance compared with baseline? [3] shows that training with more steps can help improve performance. Does this phenomenon still remain in your method?




[1] Clark, Kevin, et al. "Electra: Pre-training text encoders as discriminators rather than generators." ICLR. 2020.

[2] Gong, Linyuan, et al. "Efficient training of bert by progressively stacking." ICML. 2019.

[3] Liu, Yinhan, et al. "Roberta: A robustly optimized bert pretraining approach." arXiv preprint arXiv:1907.11692 (2019).

---

> ### Author Response · Authors · 2020-11-23
> **Thank you for reviewing this paper and giving your useful comments.**
>
> 1. The most related method to this paper is [2]. Thus, it is better to give more comparisons and discussions in the Experiment, so that the author can know the advantages of MSLT compared with [2].
>
> Response: Thank you for your advice. We have compared the proposed method with the progressively stacking method [2] in the revised version
>
> 2. From Figure 4, it seems that BERT-base (baseline) needs to train around 80 hours. If training with MSLT using the same time as baselines (like train 2M steps), how about the performance compared with baseline? [3] shows that training with more steps can help improve performance. Does this phenomenon still remain in your method
>
> Response: Thank you for you advice. We have compared our method with the baseline using the same training time.  However, we control each method uses 40 hours in total. We showd that in the same (limited) time, our method can achieve much better performance.  However, if we want to see the performance of the models trained using 2M steps, we need to train the model from scratch. Because we need to allocate suitable ratio of steps for each stage. So the existing checkpoint can not be used.  It is too expensive for us to retrain a BERT-Base and BERT-Large model with 2M steps and it is difficult for us to finish this task in this short period of the rebuttal time, since we need to do some other experiments to evaluate our method.

---

### Official Review · AnonReviewer4 · 2020-10-27
**Clear paper but lack of significance**

**Rating:** 5
**Confidence:** 5

**Review:**

This paper presents a training strategy to progressively adding top transformer layers, which results in training time speedup. Usually when training transformers, all layers are updated simultaneously. The author found out doing a multi-stage layer-wise schedule helps convergence speed without significant performance degradation.

The paper is written very clearly. The idea is conveyed with sufficient background and related work.
My biggest concern is from the originality and significance. It looks to me the approach that the paper proposed is a very straightforward extension from the work of (Gong et al., 2019).  Instead of updating all parameters in all transformer layers, this paper freezes bottom layers and only update top transformer layer (newly added). With experiments performed on base and large BERT models on GLUE dataset, this training strategy is approved to have slightly drop of quality but faster convergence speed.

I would argue that this is a great investigation and experimentation but it does not meet the criteria of acceptance. In section 3.1, the author has used the attention distribution (before and after) to motivate the work in this paper. However, if the author could discuss/explains why this change affects the convergence speed in analytical/mathematical solution, that would definitely gives more credit to originality and significance.  I would be more than willing to re-evaluate my ratings if that happens.

Minor comments/typos:

* abstract: "have has achieved" -> "have achieved".

---

> ### Author Response · Authors · 2020-11-23
> **Thank you for reviewing this paper and giving your useful comments.**
>
> 1. I would argue that this is a great investigation and experimentation but it does not meet the criteria of acceptance. In section 3.1, the author has used the attention distribution (before and after) to motivate the work in this paper. However, if the author could discuss/explains why this change affects the convergence speed in analytical/mathematical solution, that would definitely gives more credit to originality and significance. I would be more than willing to re-evaluate my ratings if that happens
>
> Response: In the revised version, we have added a discussion setion (section 5) to explain why the proposed method can speed up the training of BERT. The main reason is that we adopt the alternating optimization technique in our training strategy. The alternating optimization method is very suitable for dealing with the multi-variable optimization problem in which updating a part of variables is much easier than updating all variables. While the network training problem just fits such a case.
>
> 2. abstract: "have has achieved" -> "have achieved".
>
> Response: Thank you for pointing out this. We have modified it and we have checked the whole paper carefully.

---

### Official Review · AnonReviewer1 · 2020-10-28
**Comparison with ALBERT**

**Rating:** 5
**Confidence:** 4

**Review:**

The authors propose a multi-stage layerwise training (MSLT) approach to reduce the training time of BERT.   Experimental results show that the proposed method can achieve  110%+ training speedup without significant performance degradation.

Overall the idea of multi-stage layerwise training is reasonable and the results look promising.

My major concern about the work is the empirical comparisons. ALBERT, a closely related work, is not compared in this work.  This paper claims that "ALBERT has almost the same computational complexity as BERT, training an ALBERT model is still very time-consuming."
I checked the ALBERT paper, which claims 1.7x training speedup.
"An ALBERT configuration similar to BERT-large has 18x fewer parameters and can be trained about 1.7x faster. The parameter reduction techniques also act as a form of regularization that stabilizes the training and helps with generalization."
Seems the proposed method in this work (with 1.1x speedup) is not as good as ALBERT. I suggest to take ALBERT as backbone and check whether the MSLT can speed up the training of ALBERT.

Besides, it is better to also test on more complex downstream tasks, such as SQuAD1.1/2.0 and RACE.

---

> ### Author Response · Authors · 2020-11-11
> **Clarifying the difference of 1.7x speedup in ALBERT and 110%+ speedup in this paper**
>
> Thank you for reviewing this paper and giving your useful comments.
>
> We are sorry that some expression was not clear enough. In the ALBERT paper (according to the Table 2 in ALBERT), 1.7x speedup means the training speed of ALBERT is 1.7 times of BERT, while in this paper, 110%+ speedup means the training speed of our method is 2.1 times of original BERT.  In other words, ALBERT reduces about 42% training time and our method reduces about 55% training time. In addition, from Table 2 in ALBERT, we can see that, though 1.7x speedup is achieved, the performance of ALBERT-large is much worse than that of BERT-large (83.5 vs 86.6 in MNLI). The performance of ALBERT-xlarge  is similar to that of BERT-large (86.4 vs 86.6 in MNLI), but training ALBERT-xlarge requires even more time than training BERT-large.  In contrast, our method can achieve 2.1x speedup without significant performance degradation (86.2 vs 86.5 in MNLI).  Hence, our method is much more time efficient than ALBERT. We think the main advantage of ALBERT is its high memory efficiency.
>
> We will express this part more clearly and show more downstream task results to verify the effectiveness of our method in next version.

---

> > ### Comment · AnonReviewer1 · 2020-11-23
> > **ALBERT and ELECTRA**
> >
> > Thanks for the clarification. It largely addresses my concerns on the comparison with ALBERT. I double-checked the ALBERT paper. It indeed does not target at training speedup; instead, it focuses on parameter reduction (i.e., using fewer parameters).
> >
> > A closely related work is ELECTRA, which also focuses on training speedup and achieves 10x speedup over ALBERT with better accuracy (see Table 3 in the paper). The method in this work should be compared with ELECTRA rather than ALBERT. Seems ELECTRA is much faster than this work, 10x vs 1.25x speedup over ALBERT.

---

> > > ### Author Response · Authors · 2020-11-23
> > > **Response to reviewer 1**
> > >
> > > 1.Thanks for your comment. Generally, ELECTRA is a much more powerful model compared with BERT and ALBERT. ELECTRA introduces an effective task named replaced token detection (RTD) for pretraining, which can significantly improve the performance. However, if training with the same training steps, according to the ELECTRA paper, we can see that ELECTRA even requires more training time than BERT, with the same  hyperparameters.  ELECTRA does not make each step training more faster. It generates more effective samples to make the model stronger. Indeed, to some extent, improving the performance is to improve the speed. For example, using the sentence-order prediction (SOP) loss instead of the NSP loss, the performance of the model can also be improved. However, the goal of this paper is to provide a general efficient training strategy, and our method is orthogonal to other methods, namely it can be used in conjunction with the other techniques, including adopting more effective pretraining tasks like ELECTRA.  We choose the BERT model as the representative in this paper because it is the most popular in recent years and it is the basic of the later proposed models. However, our method can also be used to speed up the training of the other BERT based models. In the revised version, we have shown that the proposed method can also speed up the training of ALBERT.  We believe that the proposed method can also be combined with ELECTRA. However, we do not have enough time to perform new experiment and we also do not have more space to show more results in the paper currently.
> > >
> > > 2. The results on SQuAD tasks are added in the revised version.

---

### Official Review · AnonReviewer3 · 2020-10-29
**Interesting and neat idea but the experiments are limited**

**Rating:** 6
**Confidence:** 4

**Review:**

The work proposes a simple enough idea to speed up the training of BERT by progressively stacking new layers while fixing older layers. Empirically, with the same number of training steps (and less time), the proposed method can achieve a comparable performance to the original BERT. When the same amount of running time (more steps) is used, the proposed strategy can further improve the performance.

One problem with the current paper is the empirical evaluation is only conducted on the GLUE benchmark, which is sequence-level and relatively simple. I think an experience on a slightly more difficult task such as SQuAD, which also requires token-level prediction, would be necessary to test the capacity boundary of the proposed approach.

Another question is what would happen or what the performance would be if the entire model is not jointly trained for the last 20% steps. This information will help to better understand this method.

In addition, the original motivation of the work comes from the fact that the attention patterns in the bottom layers do not change much after jointly trained with more higher layers. However, this does not mean the lower-layer attention patterns don't change much if the entire network is jointly trained from scratch. To truly establish the validity of motivation, it would be good to monitor and evaluate how much the lower-layer attention patterns change when jointly trained.

---

> ### Author Response · Authors · 2020-11-23
> **Thank you for reviewing our paper.**
>
> Thank you for reviewing this paper and giving your useful comments.
>
> 1. I think an experience on a slightly more difficult task such as SQuAD, which also requires token-level prediction, would be necessary to test the capacity boundary of the proposed approach.
>
> Response: We have added the SQuAD tasks in the revised version to better evaluate the proposed method.
>
> 2. What would happen or what the performance would be if the entire model is not jointly trained for the last 20% steps. This information will help to better understand this method.
>
> Response:  We have added a subsection (section 4.5) in the Experiment section to investigate the effect of  jointly retraining in the revised version.
>
> 3. The original motivation of the work comes from the fact that the attention patterns in the bottom layers do not change much after jointly trained with more higher layers. However, this does not mean the lower-layer attention patterns don't change much if the entire network is jointly trained from scratch. To truly establish the validity of motivation, it would be good to monitor and evaluate how much the lower-layer attention patterns change when jointly trained.
>
> Response: Thank you for pointing out this. We have added the attention distribution of the original BERT model trained from scratch in section 3.1 of the revised version.

---

### Author Response · Authors · 2020-11-23
**New version submitted**

We have submitted a new version of the paper. The main modifications are listed below:
1. We add the SQuAD tasks for testing the proposed method. 2. We make a discussion on why the proposed method can speed up training. 3. We apply the proposed method to ALBERT to speed up its training. 4. We add an ablation test to show the effect of the retraining stage.

---

### Decision · Program_Chairs · 2021-01-07
**Final Decision**

**Decision:**

Reject

**Comment:**

This paper proposed a variant of the existing training method "progressive stacking", and showed good empirical results comparing with the normal training procedure for BERT models. It contains some interesting points on the technical side, e.g, freezing the bottom layers when training newly added layers, but there are several concerns:  (1) This proposed method is called progressive stacking 2.0 but there is no comparison against the original progressive stacking in experiments. We had to check the empirical results in the original paper of progressive stacking, and did not notice any performance improvement of this new method over the original one; (2) The proposed method even introduced one more hyperparameter to tune: the number of top layers to copy.  This hyperparameter seems hard to choose. Different choices of this hyperparameter may dramatically impact the performance of this new method.  An ablation study on this should be conducted, e.g.,  what will the results look like if we only copy the last layer?  (3) The original progressive stacking method does not provide any practical guidance on how to determine the time to stack when the training goes. This severely limits the practical value of progressive stacking. If one stacks layers too early or too late, the stacking method may have no advantage at all. Unfortunately, this method called 2.0 still leaves this most critical issue away.